# An endothelial regulatory module links blood pressure regulation with elite athletic performance

Kim Fegraeus[1]⊙*, Maria K. Rosengren[2]⊙, Rakan Naboulsi[2,3], Ludovic Orlando[4], Magnus Åbrink[5], Ahmad Jouni[2], Brandon D. Velie[6], Amanda Raine[1], Beate Egner[7], C Mikael Mattsson[8], Karin Lång[9], Artemy Zhigulev[10], Hanna M. Björck[9], Anders Franco-Cereceda[11], Per Eriksson[9], Göran Andersson[2], Pelin Sahlén[10], Jennifer R. S. Meadows[12]‡, Gabriella Lindgren[2,13]‡*

1 Department of Medical Sciences, Science for life laboratory, Uppsala University, Sweden, 2 Department of Animal Biosciences, Swedish University of Agricultural Sciences Uppsala, Sweden, 3 Childhood Cancer Research Unit, Department of Women's and Children's Health, Karolinska Institute, Stockholm, 4 Centre d'Anthropobiologie et de Génomique de Toulouse (CNRS UMR 5288), Université Paul Sabatier, Toulouse, France, 5 Department of Biomedical Sciences and Veterinary Public Health, Swedish University of Agricultural Sciences, Uppsala, Sweden, 6 School of Life & Environmental Sciences, University of Sydney, Sydney, Australia, 7 Department of Cardio-Vascular Research, Veterinary Academy of Higher Learning, Babenhausen, Germany, 8 Silicon Valley Exercise Analytics (svexa), MenloPark, CA, United States of America, 9 Division of Cardiovascular Medicine, Center for Molecular Medicine, Department of Medicine, Karolinska Institutet, Stockholm, Karolinska University Hospital, Solna, Sweden, 10 KTH Royal Institute of Technology, School of Chemistry, Biotechnology and Health, Science for Life Laboratory, Stockholm, Sweden, 11 Section of Cardiothoracic Surgery, Department of Molecular Medicine and Surgery, Karolinska Institutet, Stockholm, Sweden, 12 Department of Medical Biochemistry and Microbiology, Science for Life Laboratory, Uppsala University, Uppsala, Sweden, 13 Center for Animal Breeding and Genetics, Department of Biosystems, KU Leuven, Leuven, Belgium

⊙ These authors contributed equally to this work.
‡ These authors are shared senior authors on this work.
* Kim_Matilda@hotmail.com (KF); gabriella.lindgren@slu.se (GL)

**Data Availability Statement:** The racing performance data that support the findings of this study are available from the Swedish Trotting

## Abstract

The control of transcription is crucial for homeostasis in mammals. A previous selective sweep analysis of horse racing performance revealed a 19.6 kb candidate regulatory region 50 kb downstream of the Endothelin3 (*EDN3*) gene. Here, the region was narrowed to a 5.5 kb span of 14 SNVs, with elite and sub-elite haplotypes analyzed for association to racing performance, blood pressure and plasma levels of EDN3 in Coldblooded trotters and Standardbreds. Comparative analysis of human HiCap data identified the span as an enhancer cluster active in endothelial cells, interacting with genes relevant to blood pressure regulation. Coldblooded trotters with the sub-elite haplotype had significantly higher blood pressure compared to horses with the elite performing haplotype during exercise. Alleles within the elite haplotype were part of the standing variation in pre-domestication horses, and have risen in frequency during the era of breed development and selection. These results advance our understanding of the molecular genetics of athletic performance and vascular traits in both horses and humans.

Association (Stockholm, Sweden), but restrictions apply to the availability of these data, which were used under license for the current study, and so are not publicly available. However, data are available from The Swedish Trotting Association (Svensk Travsport, Box 201 51, 161 02 Bromma, Sweden). The contact person is Christina Olsson, head of the breeding department at The Swedish Trotting Association, kundtjanst@travsport.se. The WGS data has been deposited in the BioProject database (NCBI repository SRA) with ID PRJNA1045044 according to their guidelines. According to the ethical permit related to the study (application number 2006/784-31/1 and 2012/1633-31/4) approved by the Human Research Ethics Committee at Karolinska Stockholm, Sweden, the sequencing data from living patients cannot be shared due to patient privacy regulations. We therefore only share the summary or processed form of the sequencing data in S5–S7 Tables. More information can be found at Etikprövningsmyndigheten, Box 2110, 750 02, Uppsala, or registrator@etikprovning.se.

**Funding:** The Swedish Research Council for Environment, Agricultural Sciences and Spatial Planning (FORMAS) (to GL) and The Swedish Research Council (VR) (to GL). This project has also received funding from the CNRS, University Paul Sabatier (AnimalFarm IRP) (to LO), and the European Research Council (ERC) under the European Union's Horizon 2020 research and innovation program (grant agreement 681605-PEGASUS) (to LO). K.F. and A.R. were supported by grants from FORMAS (2020-01135) (to AR). K. F. received salary from FORMAS (2020-01135). The funders had no role in study design, data collection and analysis, decision to publish, or preparation of the manuscript.

**Competing interests:** The authors have declared that no competing interests exist.

## Author summary

A previous study discovered that a genomic region close to the Endothelin3 gene was associated with harness racing performance. Here, careful phenotypic documentation of athletic performance and blood pressure measurements in horses, followed by state-of-the-art genomics, allowed us to identify a 5.5 kb regulatory region located approximately 50 kb 3' of the *EDN3* gene. A comparative analysis of the region using human HiCap data supported a regulatory role as, in endothelial cells, interaction was observed between the region and multiple genes relevant to blood pressure regulation and athletic performance. Long range cis-regulatory modules are critical for cooperatively controlling multiple genes located within transcriptionally active domains. We measured blood pressure in Coldblooded trotters during exercise and demonstrated that horses with two copies of the elite-performing haplotype had lower blood pressure during exercise and better racing performance results, compared to horses with two copies of the sub-elite performing haplotype. In addition, horses with the elite-performing haplotype also had higher levels of Endothelin3 in plasma. The results reported here are important for understanding the biological mechanisms behind blood pressure regulation in relation to racing performance in both horses and humans.

## Background

Despite the apparent potential of domestic animal models to provide valuable insights into the natural biological mechanisms driving human traits and disease, their use to date has been limited. In the past, the use of domestic animals as models for genomic research has provided basic knowledge concerning gene function and biological mechanisms and a complementary view on genotype–phenotype relationships [1]. Recent advances in the availability and quality of human and mammalian reference genomes, plus the technological advances required for their alignment, are revealing both coding and non-coding conserved bases, key to unraveling shared gene function and regulation [2,3]. The horse is one of the most popular species for studying athletic performance. They have been intensively selected for centuries, to encompass the optimal physical capacity for strength, speed, and endurance [4,5]. Additionally, their recent population history, involving closed populations selected for similar phenotypic traits within breeds and large variations across breeds, has created a favorable genome structure for genetic mapping. These factors combined make the horse an optimal model for studying the molecular genetics underlying athletic performance and the complex biological processes activated by exercise [1,4,6,7]. Previous research in horses has begun to unravel the genetics of complex traits, such as muscle mass and locomotion patterns, on athletic performance, with many potential candidate genetic variants identified [8–14]. However, it is much more challenging to understand the mechanisms by which the identified variants exert their functions.

We have previously used a unique, three breed admixture Nordic horse model, to study the genetics of athletic performance traits [15,16]. The basis for this model is that racing Cold-blooded trotters (CBTs) predominantly originate from the North Swedish Draught horses (NSDs), a sturdy breed used in farming and forestry. It is well established that some cross-breeding occurred between Standardbreds (SBs) (the most commonly used breed for harness racing) and CBTs before obligatory paternity testing was introduced in Sweden in the 1960s. However, a remarkable improvement in the racing performance of the CBT has occurred during the last fifty years. In part, it is likely that this improvement could be explained by cross-breeding, with a marked increase of favorable genetic variants originating from SBs and

introduced to the CBTs. This process should leave "genetic footprints" in the genome of CBT in the form of chromosome segments originating from SBs. We have identified such a footprint using pair-wise allele frequency distribution and selective sweep mapping [15]. Notably, our model, which is solely based on the hybrid origin of the CBT, provides a unique opportunity to study genes influencing body constitution and complex morphological traits of importance for racing success.

The harness racing selective sweep study revealed a 19.6 kb genomic region on chromosome 22, located approximately 50 kb downstream of the *Endothelin 3* (*EDN3*) gene, under selection in CBT [15]. Five SNVs in high linkage disequilibrium (LD, $r^2$ = 0.92–0.94) were significantly associated with racing performance, including the number of victories, earnings, and racing times. Variant rs69244086 C>T (EquCab3.0, ECA22:46717860) showed the strongest association within the sweep region, and was further genotyped in 18 additional horse breeds. The favorable T allele was found in high frequency in breeds used for racing, while it generally remained at low frequency in ponies and draught horses. We hypothesized that the identified region might contain a regulatory element influencing either the expression of *EDN3* or other genes nearby [15].

Genomic regulatory regions, such as putative enhancers, are difficult to study simply because they are hard to characterize [17]. They can be cell type and condition specific, acting both upstream and downstream of target genes, or over long genomic distances, where genome looping brings them into proximity to their target genes [18]. Targeted chromosome conformation capture (HiCap) is an experimental method that can detect genomic loops mediating regulatory interactions between such regions and the promoters of their target genes [19,20]. In this study, we fine-mapped the harness racing selective sweep, and used comparative data to interrogate its potential function. In addition to equine epigenetic data, we have used HiCap datasets produced using twelve different human cell types, including vascular endothelial cells, to indicate a role of our potential regulatory region and the target genes involved. Further, since the sweep region flanks the *GNAS-EDN3* region identified in multiple human blood pressure GWAS [21,22], we evaluated the effect of the minimal sweep horse haplotypes on blood pressure, as well as EDN1 and EDN3 plasma levels, before and during exercise. Finally, key genetic variants within the sweep region were characterized for spatial and temporal distribution within past and present horse populations. Taken together, the results presented here indicate the identification of a regulatory unit, likely important for vascular traits and performance in horses and humans.

## Results

### Fine-mapping and comparative analyses suggest a regulatory role of the selected region

With a four-step process, we reduced the 19.6 kb sweep region from the previous study [15] to a minimum shared 5.5 kb region (Fig 1A and 1B). In step one, we repeated the racing performance association analysis using 251 more horse samples (total n = 629) with SNV genotypes extracted from the 670K Axiom Equine Genotyping Array. However, supplementing the original 378 CBTs with additional samples did not narrow the 19.6 kb region [15]. Seven SNVs within the sweep (five significantly associated with racing performance and two flanking SNVs) [15] were extracted from the array, including the most significant one: rs69244086 [15]. Pairwise LD centered on the previously most associated SNV, rs69244086 [15], resulted in the same core five SNV region (ECA22:46,717,451–46,718,964, $r^2$ > 0.6), and analysis of linear models showed that these SNVs remained significantly associated with harness racing performance traits (i.e., number of victories, earnings, and race times, P ≤ 0.05) (S1 Table).

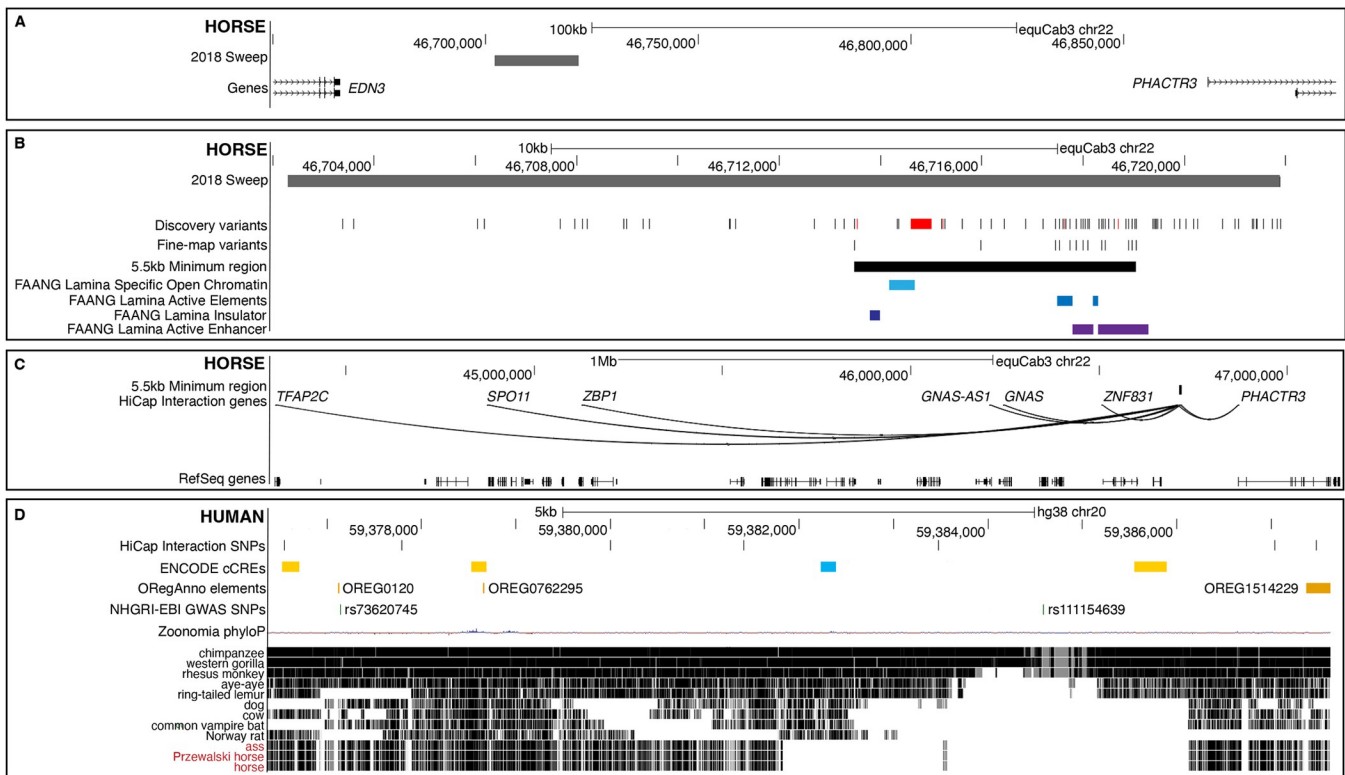

**Fig 1. Sweep fine-mapping and functional exploration.** A) Location of the 2018 selective sweep. B) Fine-mapped region including discovery variants, fine-mapped SNVs associated with racing performance, the refined minimum sweep region as well as active elements in lamina from FAANG data. Discovery SNVs are indicated in black, deletions in red. C) HiCap interacting SNV and genes, lifted from humans. D) The span of the minimum sweep region covers an extended 11 kb space in the human reference genome, hg38. HiCap interacting SNVs are indicated in relation to known regulatory elements and SNVs reported from GWAS studies. Zoonomia phyloP scores and cactus alignments show the conservation of this region across mammals. The yellow boxes represent distal enhancer-like signatures and the blue box represent a CTCF-only.

In step two, we performed whole genome sequencing (WGS) of two CBTs and two SBs to increase variant density across the 19.6 kb sweep region (Fig 1B). In each breed, horses were selected to be homozygous for different alleles at SNV rs69244086 (CC and TT, respectively) and had either high (TT) or low (CC) earnings per start (S2 Table). From the 19.6 kb sweep region, 78 SNVs and six indels (one 400 bp deletion, one 23 bp deletion and four single bp INDELs) were identified (Fig 1B and S3 Table).

In step three, we prioritized variants for further analysis. The 400 bp deletion (ECA22:46,714,602–46,715,003) was absent in horses with low earnings per start and either heterozygous (SB) or homozygous (CBT) in elite performing horses. However, this variant was not significantly associated with racing performance traits following genotyping and analyses in a further 497 CBTs (linear models, S4 Table). Due to technology constraints, 25 of the available 82 single base pair variants were selected for genotyping in a larger horse material. Variant selection was based on available pooled allele frequency data matching admixture expectation in CBTs, SBs and NSD [16], variant spacing across the region, and success in MassArray design (see Methods). The 24 SNVs and one single base pair deletion (Fig 1B) were genotyped in 391 horses sampled from 12 different breeds (including 210 CBTs), as well as three Przewalski horses. SNV rs69244086 was not included in this set, but was either directly genotyped or imputed across the dataset. After quality control, 24 SNVs and 394 horses (including the Pzrewalskis) were available for further analysis. Pairwise LD calculations with SNV

**Table 1. Haplotype frequencies in different breeds.**

| Haplotype (H)[1] | All breeds (n = 394) | CBT (n = 210) | Arabians (n = 29) | TB (n = 30) | SB (n = 45) | Ardennes (n = 20) | NSD (n = 18) | Exmoor (n = 23) | Icelandic (n = 6) |
|---|---|---|---|---|---|---|---|---|---|
| H1) AAGCGTTTCTCAAA | 0.58 | 0.62 | 1.00 | 0.93 | 0.77 | 0.13 | 0.06 | 0.00 | 0.00 |
| H2) GGTTACCCTCTGGG | 0.41 | 0.38 | 0.00 | 0.07 | 0.23 | 0.78 | 0.92 | 1.00 | 0.75 |
| H3) AAGCGTTTCTCGAA | <0.01 | 0.00 | 0.00 | 0.00 | 0.00 | 0.07 | 0.00 | 0.00 | 0.00 |
| H4) AAGCGTTTCTCAAG | <0.01 | 0.00 | 0.00 | 0.00 | 0.00 | 0.00 | 0.00 | 0.00 | 0.09 |
| H5) AAGTGTTTCTCAAG | <0.01 | 0.00 | 0.00 | 0.00 | 0.00 | 0.00 | 0.00 | 0.00 | 0.08 |
| H6) GGGCGTCCCTCAAA | <0.01 | 0.00 | 0.00 | 0.00 | 0.00 | 0.00 | 0.00 | 0.00 | 0.04 |
| H7) GGGCGTTCCTCAAA | <0.01 | 0.00 | 0.00 | 0.00 | 0.00 | 0.00 | 0.00 | 0.00 | 0.05 |

[1] Order of SNVs, rs395117226, rs397265747, rs396474304, rs69244081, rs69244084, rs69244085, rs69244086, rs69244088, rs69244089, rs396281591, rs394573286, rs69244091, rs69244093, rs69244095.

CBT: Coldblooded trotter, TB: Thoroughbred, SB: Standardbred, NSD: North-Swedish draught horse

rs69244086 revealed 15 SNVs in strong LD (r2 ≥ 0.53, 14 SNVs with r2 > 0.98) (ECA22:46,708,983–46,719,042). LD decayed outside the region ($r^2 < 0.07$), leaving 14 SNVs for haplotype analysis (Fig 1B). Phasing revealed in total 14 different haplotypes, with seven haplotypes found at a frequency > 2% in the total sample set, or within each breed represented by five or more individuals (Table 1). Generalized linear model (GLM) regression in the R environment (23) was used to test the association between the two haplotypes harbored by CBTs (n = 165) and SBs (n = 38), and harness racing performance traits. For both CBTs and SBs, the H1 haplotype, carrying the rs69244086-T high-performance associated allele, was the most common (Table 2). In both breeds, the H2 haplotype demonstrated a significantly negative effect on all performance traits tested, except for the number of starts in SBs (P = 0.59) and the number of wins in SBs (P = 0.06) (Table 2). The above analysis showed that the minimum 5,564 bp, 14 SNV shared haplotype (ECA22: 46,713,478–46,719,042), significantly associated with racing performance in both CBTs and SBs. This allowed for the definition of H1 as an elite-performing haplotype (EPH: AAGCGTTTCTCAAA) and H2 as a sub-elite-performing haplotype (SPH: GGTTACCCTCTGGG). Of note, the reference EquCab3.0 haploid reference genome, generated from a Thoroughbred, represents the SPH haplotype.

In step four, we assessed the potential functional impact of the minimum shared 5.5 kb region using comparative data from both horse and human resources (Fig 1C and 1D). Epigenetic data drawn from nine tissues was generated by the Equine section of Functional Annotation of Animal Genomes (FAANG) [24] for two Thoroughbreds, both homozygous for the EPH haplotype. No clear signals were observed in the minimal region for four marks (H3K27ac, H3K27me3, H3K4me1, H3K4me3) measured across eight different tissues (adipose, brain, heart, liver, lung, muscle, ovary, skin). But in lamina we observed an insulator and an active enhancer in our 5.5 kb minimum shared region (Fig 1). To access additional functional data points, we lifted the minimum shared 5.5 kb region and contained SNVs to the human reference genome, hg38. In comparative analyses, the liftover indicated that, although not well conserved, the region had regulatory potential, including Open Regulatory Annotation database (ORegAnno) elements [25], ENCODE cCREs [18] and GTEx cis-eQTL variants

**Table 2. Haplotype frequencies, haplotype coefficient and P-values for the performance analysis in Coldblooded trotters (n = 165) and Standardbreds (n = 38).**

| | Coldblooded trotters | | | | Standardbreds | | | |
|---|---|---|---|---|---|---|---|---|
| | Haplotype H1 | | Haplotype H2 | | Haplotype H1 | | Haplotype H2 | |
| Haplotype[1] | AAGCGTTTCTCAAA | | GGTTACCCTCTGGG | | AAGCGTTTCTCAAA | | GGTTACCCTCTGGG | |
| Frequency | 0.61 | | 0.39 | | 0.78 | | 0.22 | |
| | Haplotype coefficient[2] | P-value[3] | Haplotype coefficient | P-value[3] | Haplotype coefficient | P-value[3] | Haplotype coefficient | P-value[3] |
| No. of starts[5] | - | - | -1.02 | **<0.001** | - | - | 0.25 | 0.59 |
| No. of wins[5] | - | - | -0.12 | **<0.001** | - | - | -0.12 | 0.06 |
| No. of placings[5] | - | - | -0.10 | **<0.001** | - | - | -0.12 | **0.01** |
| Earnings (SEK)[4,5] | - | - | -0.39 | **<0.001** | - | - | -1.30 | **<0.001** |
| Earnings/start[6] | - | - | -0.37 | **<0.001** | - | - | -1.08 | **<0.001** |
| Best time (sec/km)[5,6] | - | - | 0.05 | **<0.001** | - | - | 0.31 | **0.001** |

[1] Order of SNVs. rs395117226, rs397265747, rs396474304, rs69244081, rs69244084, rs69244085, rs69244086, rs69244088, rs69244089, rs396281591, rs394573286, rs69244091, rs69244093, rs69244095

[2] Haplotype coefficient, or effect size. This is the effect of each haplotype compared to the base haplotype (H1), i.e., one copy of the H2 haplotype will reduce the square root transformed number of starts with ~1.

[3] A GLM regression analysis was performed in R. Sex, age, birth country, number of starts (when applicable) and DMRT3 genotype (only in CBTs) were included as fixed effects. Significant results (P ≤ 0.05) in bold

[4] SEK, Swedish kronor

[5] Log transformed values (log10 +1) were used for wins and placings. Number of starts were square root transformed. Earnings and earnings per start were transformed applying ln(earnings + 1000)

[6] 173 CBTs and 36 SBs were included in the analysis of best racing time

regulating *EDN3* in esophagus mucosa tissue [26] (Fig 1D). We further investigated the regulatory potential of the minimum shared 5.5 kb region using comparative chromatin interaction profiles in 12 different human cell types, including iPS cells and a neural cell line (see Methods and S5 Table). SNVs lifted to the minimum 5.5 kb region showed interactions with the promoters of multiple genes, but only in the vascular endothelial cell datasets (Fig 1C and 1D and S5 Table). With a relaxed threshold, requiring support evidence from at least two samples, direct interaction between human to horse lifted SNVs and seven genes were observed (Fig 1C). These interactions included the proximal cell membrane-cytoskeleton dynamics gene, *PHACTR3* [27], through to the megabase distal cell reprogramming transcription factor gene, *TFAP2C* [28]. While specific SNVs in humans are unlikely to have the exact same role in horses, we never-the-less explored the human direct, and indirect, interaction network to gain a wider perspective of genes which may be regulated by the minimum shared 5.5 kb region (Fig 2). Here, support was required from five samples, and all human loci are considered, even if these are not currently supported with horse annotation evidence. We see that the 5.5 kb region (putative regulatory locus, PutRegLocus) interacts with the *EDN3* promoter indirectly, via *GNAS* and two other promoters (Fig 2). GWAS variant rs16982520, located within *ZNF831*, is associated with hypertension [29,30], systolic blood pressure [31] and mean arterial pressure [32] and interacts directly with *EDN3* and four direct interactors of the putative regulatory region (*GNAS*, *RP4-614C15.2*, *SPO11* and *ZNF831*), suggesting a complex regulatory pattern. In total, there were 15 protein-coding genes that directly or indirectly interacted with the putative regulatory region (S6 Table). We performed a gene enrichment analysis for these 15 genes and found that several human phenotype terms related to thyroid hormone regulation were enriched (S7 Table).

Moving back to the horse genome, we investigated the *in silico* regulatory potential of each allele from the 14 SNVs associated with racing performance and included in the minimum

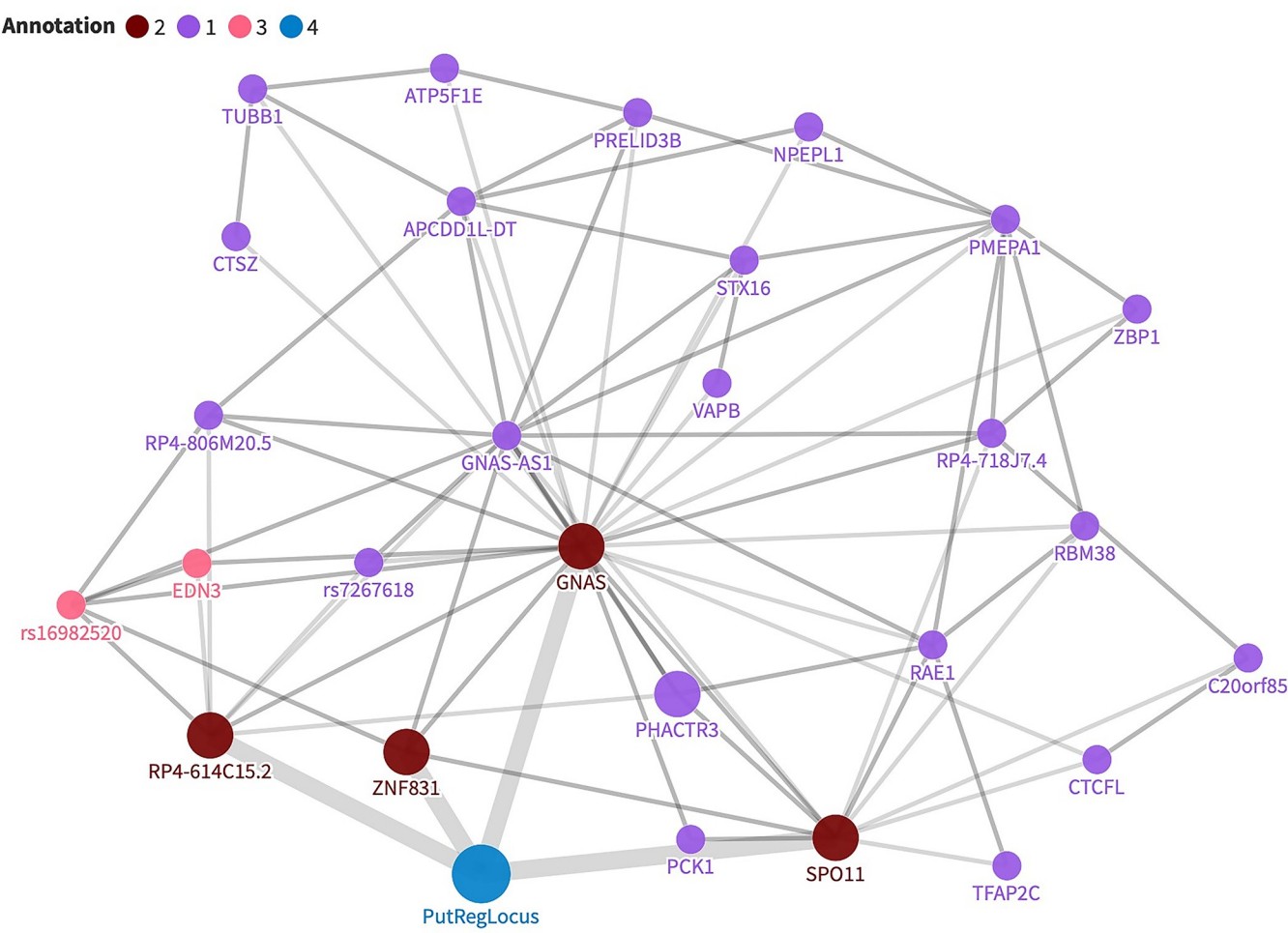

**Fig 2. The direct and indirect interactions of the minimum haplotype lifted over to hg38 (PutRegLocus, blue) in vascular endothelial cells.** There were four direct interactors of the locus (marked in dark red). Some of the direct interactors of the PutRegLocus interacted with EDN3 promoter and rs16982520 variant (pink). The direct interactors of PutRegLocus are connected with thick edges.

shared 5.5 kb region, to alter transcription factor binding. Using the EquCab3 genome reference and the Homo sapiens Comprehensive Model Collection (HOCOMOCO) transcription factor binding model database [33] in motifbreakR [34], we found that all alleles had the potential to cause alterations of high effect. SNVs rs69244086 C>T and rs69244089 T>C are illustrated as examples in S1 Fig. In S8 Table, the transcription factor binding scores for all analyzed variable sites are listed.

### The elite performing haplotype is significantly associated with improved racing performance

Given that all 14 associated variants within the minimum shared 5.5 kb locus were in perfect LD in CBTs and SBs, we used the genotypes at rs69244089 as a proxy for EPH (C allele) and SPH (T allele) haplotypic pairs and re-assessed the association between haplotypes and racing performance. For CBTs (n = 516), EPH homo- or heterozygotes outperformed the SPH homozygotes (Table 3). In SBs, there were no statistically significant differences between the three genotypes (Table 4). However, when analyzing horses carrying at least one T allele versus CC horses, the results were significant for several performance traits (Table 4), including the

**Table 3. Performance results in 516 Coldblooded trotters for SNV rs69244089 T>C.**

| Genotype[2] | TT (n = 63) | | TC (n = 244) | | CC (n = 209) | | P[3] | | |
|---|---|---|---|---|---|---|---|---|---|
| Performance trait[1] | Mean | Median | Mean | Median | Mean | Median | TT/TC | TT/CC | TC/CC |
| No. of starts | 31.6 | 20.0 | 41.5 | 31.0 | 41.2 | 31.0 | **0.01** | **0.02** | **0.99** |
| No. of wins | 2.8 | 1.0 | 5.7 | 3.0 | 5.6 | 3.0 | **0.006** | **0.05** | 0.56 |
| No. of placings (1–3) | 9.1 | 4.0 | 14.2 | 9.0 | 13.9 | 8.0 | **0.005** | **0.04** | 0.54 |
| Earnings (SEK) | 209,486 | 51,772 | 453,084 | 148,197 | 425,448 | 116,500 | **0.009** | **0.05** | 0.65 |
| Earnings per start (SEK) | 4,760 | 2,939 | 9,666 | 4,362 | 8,188 | 3,748 | **0.003** | 0.07 | 0.26 |
| Best km time (sec)[4] | 91.1 | 91.5 | 88.5 | 88.3 | 89.0 | 89.0 | **0.002** | **0.04** | 0.37 |

[1] Log transformed values (log10 +1) were used for wins and placings. Earnings and earnings per start were transformed applying ln(earnings + 1000). Square root transformations were used for number of starts

[2] The C-allele is tagging the elite performance haplotype (EPH)

[3] A linear model was performed in R. Sex, age, birth country, number of starts and *DMRT3* genotypes were included as fixed effects (when significant). Least square means and Tukey were used as post hoc tests. Significant results (P ≤ 0.05) in bold

[4] Only races where the horse did not gallop were included

number of wins, placings and earnings. While the number of wins and placings was higher in horses with at least one T allele, the CC horses earned more money than the TT/TC horses (Table 4).

In both SBs and CBTs the rs69244089 genotypes distributed according to HWE (P = 0.16 and P = 0.35, respectively). For breeds other than CBT and SB, each individual's performance status is unknown. However, a trend for an increased frequency of the C allele in traditional performance breeds was observed, e.g., Arabian horses, Thoroughbreds, and Warmbloods. In contrast, this allele was low or absent in draft horses and ponies (Table 5).

## Increased frequency of the favorable haplotype coincides with the time of horse domestication

For all 14 SNVs within the minimum shared haplotype in the trotters, we investigated the allele frequency over time using the mapDATAge package (35) and 431 previously characterized ancient genomes [4,36–40]. For all SNVs except two, rs69244085 and rs69244086, there has

**Table 4. Performance results in 271 Standardbreds for SNV rs69244089 T>C.**

| Genotype[2] | TT (n = 7) | | TC (n = 62) | | CC (n = 202) | | P[3] | | | |
|---|---|---|---|---|---|---|---|---|---|---|
| Perf. trait[1] | Mean | Median | Mean | Median | Mean | Median | TT/TC | TT/CC | TC/CC | CC/ TT+TC |
| No. of starts | 43.0 | 48.0 | 42.9 | 38.0 | 35.6 | 31.0 | 0.85 | 0.60 | 0.52 | 0.20 |
| No. of wins | 6.7 | 5.0 | 6.3 | 4.0 | 5.7 | 5.0 | 0.82 | 0.99 | 0.10 | **0.04** |
| No. of placings (1–3) | 15.6 | 12.0 | 16.1 | 16.0 | 13.7 | 11.0 | 0.86 | 0.96 | 0.08 | **0.03** |
| Earnings (SEK) | 700,008 | 322,300 | 730,124 | 332,300 | 1,075,529 | 369,675 | 0.92 | 0.43 | 0.06 | **0.01** |
| Earnings per start (SEK) | 11,981 | 5,463 | 16,673 | 9,765 | 35,346 | 9,897 | 0.86 | 0.44 | 0.16 | **0.04** |
| Best km time (sec)[4] | 73.1 | 73.9 | 73.3 | 72.8 | 73.1 | 72.9 | 0.85 | 1.00 | 0.19 | 0.10 |

[1] Log transformed values (log10 +1) were used for wins and placings. Earnings and earnings per start were transformed applying ln(earnings + 1000). Square root transformations were used for number of starts

[2] The C-allele is tagging the elite performance haplotype (EPH)

[3] A linear model was performed in R. Sex, age, birth country and number of starts were included as fixed effects (when significant). Least square means and Tukey were used as post hoc tests. Significant results (P ≤ 0.05) in bold

[4] Only races where the horse did not gallop were included

**Table 5. Allele and genotype frequencies for SNV rs69244089 T>C.**

| Breed | n | T | C | TT | TC | CC |
|---|---|---|---|---|---|---|
| Arabian horses | 30 | 0.00 | 1.00 | 0.00 | 0.00 | 1.00 |
| Thoroughbred | 30 | 0.07 | 0.93 | 0.00 | 0.13 | 0.87 |
| Standardbred | 306 | 0.14 | 0.86 | 0.03 | 0.22 | 0.75 |
| Warmblood | 3 | 0.17 | 0.83 | 0.00 | 0.33 | 0.66 |
| Coldblooded trotters | 539 | 0.36 | 0.64 | 0.12 | 0.48 | 0.40 |
| Finnhorses | 4 | 0.50 | 0.50 | 0.50 | 0.00 | 0.50 |
| Gotland pony | 2 | 0.50 | 0.50 | 0.50 | 0.00 | 0.50 |
| Shetland pony | 18 | 0.64 | 0.36 | 0.44 | 0.39 | 0.17 |
| Ardennes | 20 | 0.78 | 0.22 | 0.65 | 0.25 | 0.10 |
| Icelandic horses | 11 | 0.86 | 0.14 | 0.73 | 0.27 | 0.00 |
| North-Swedish draught | 18 | 0.92 | 0.08 | 0.83 | 0.17 | 0.00 |
| Exmoor pony | 23 | 1.00 | 0.00 | 1.00 | 0.00 | 0.00 |
| Fjord horse | 1 | 1.00 | 0.00 | 1.00 | 0.00 | 0.00 |
| Pzrewalski | 3 | 1.00 | 0.00 | 1.00 | 0.00 | 0.00 |

been an increase of the alternate allele observed from at least 7,500 to 5,500 years ago (Fig 3). For all 14 SNVs, the two alleles were shown to segregate in specimens pre-dating the rise and spread of the DOM2 genetic lineage of modern domestic horses [35].

## Elite performance haplotype linked to lower blood pressure

To link genotype to phenotype, we assessed the physiological difference between CBTs homozygous for SPH (n = 5–7) and EPH (n = 17) before, during and post-exercise (see methods).

At rest, blood pressure measurements were also taken from eight heterozygous horses (HET). For all traits, a significant interaction was calculated between haplotype and time point. On average, horses homozygous (HOM) for SPH had significantly higher systolic blood pressure (SBP), diastolic blood pressure (DBP), and mean arterial pressure (MAP) during exercise, measured directly after the last uphill interval (Fig 4A-4C). In addition, five minutes after the uphill interval, the SPH group showed significantly higher SBP, MAP, and pulse pressure (PP) than the EPH group (Fig 4D and 4E). There were no statistically significant differences in blood pressure measurements at rest before the exercise. Although there was a higher proportion of males in the SPH group, there was no significant impact of sex on any of the blood pressure measurements. All blood pressure values are presented in S9 Table.

## Plasma concentrations of EDN1 and EDN3 differ between the haplotype groups

ELISA tests were used to measure the plasma concentration of EDN1 and EDN3 at rest and during exercise for each haplotype group. Samples with a coefficient of variation (CV) value above 20% were excluded from the analysis (S10 and S11 Tables). Horses homozygous for SPH (n = 8) had a significantly higher plasma concentration of EDN1 at rest and during exercise, compared to the other groups (Tukey´s HSD test) (Fig 5). In addition, SPH horses had lower plasma concentrations of EDN3 at rest (P = 0.06) and during training (P = 0.003) compared to horses homozygous for the EPH (Fig 5). Individual plasma values are presented in S10 and S11 Tables. There was a significant correlation between the plasma EDN3 and DPB (R = -0.66, P = 0.03) and a borderline significant correlation between EDN3 and MAP (R = -0.57, P = 0.07) during exercise. Also, there was a significant correlation between EDN1 and DBP

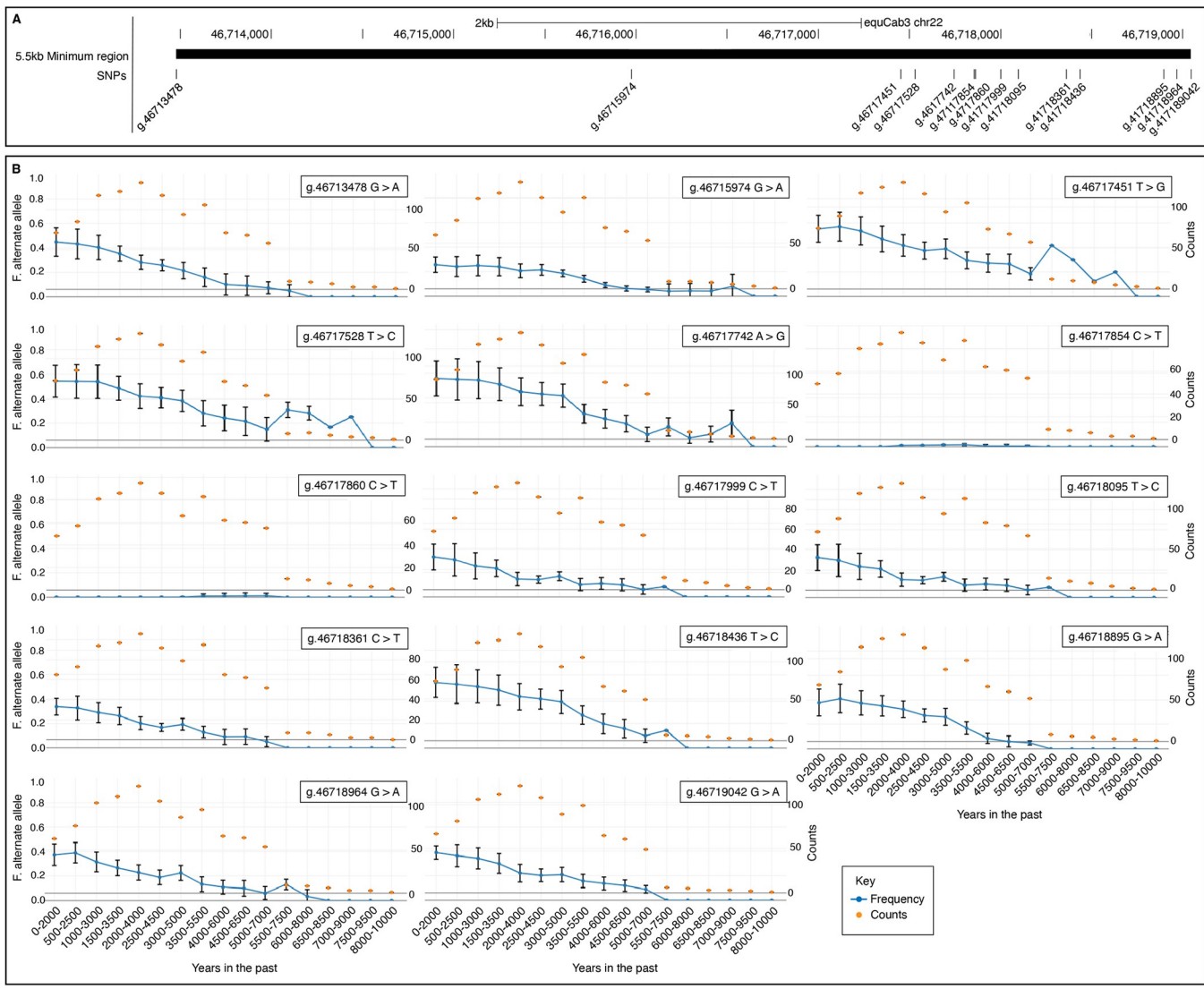

**Fig 3. Distribution of alleles over time.** A) The location of the 14 SNVs included in the minimum shared region. B) From estimated allele counts, the frequency of the alternative allele for each SNV (blue) is plotted over time in bins of 2000 years. The number of horse samples considered is also indicated (orange).

(R = 0.73, P = 0.01), SBP (R = 0.7, P = 0.02) as well as MAP (R = 0.81, P = 0.003) during exercise.

## Discussion

Naturally occurring animal models, such as livestock and companion animal species, can provide complementing views to *de novo* generated animal models, where genome editing is required and the genomic context of the mutation may be lost. Natural model populations can carry specific genetic variants that have been under artificial selection to obtain desired phenotypes. In this study, we have used such a model to study the regulatory genomics of blood pressure modulation. More fundamental knowledge on exercise related blood pressure modulation and blood pressure tuning in general is warranted, since dysregulation of these systems are linked to both cardiovascular disease and metabolic syndrome [41,42]. By

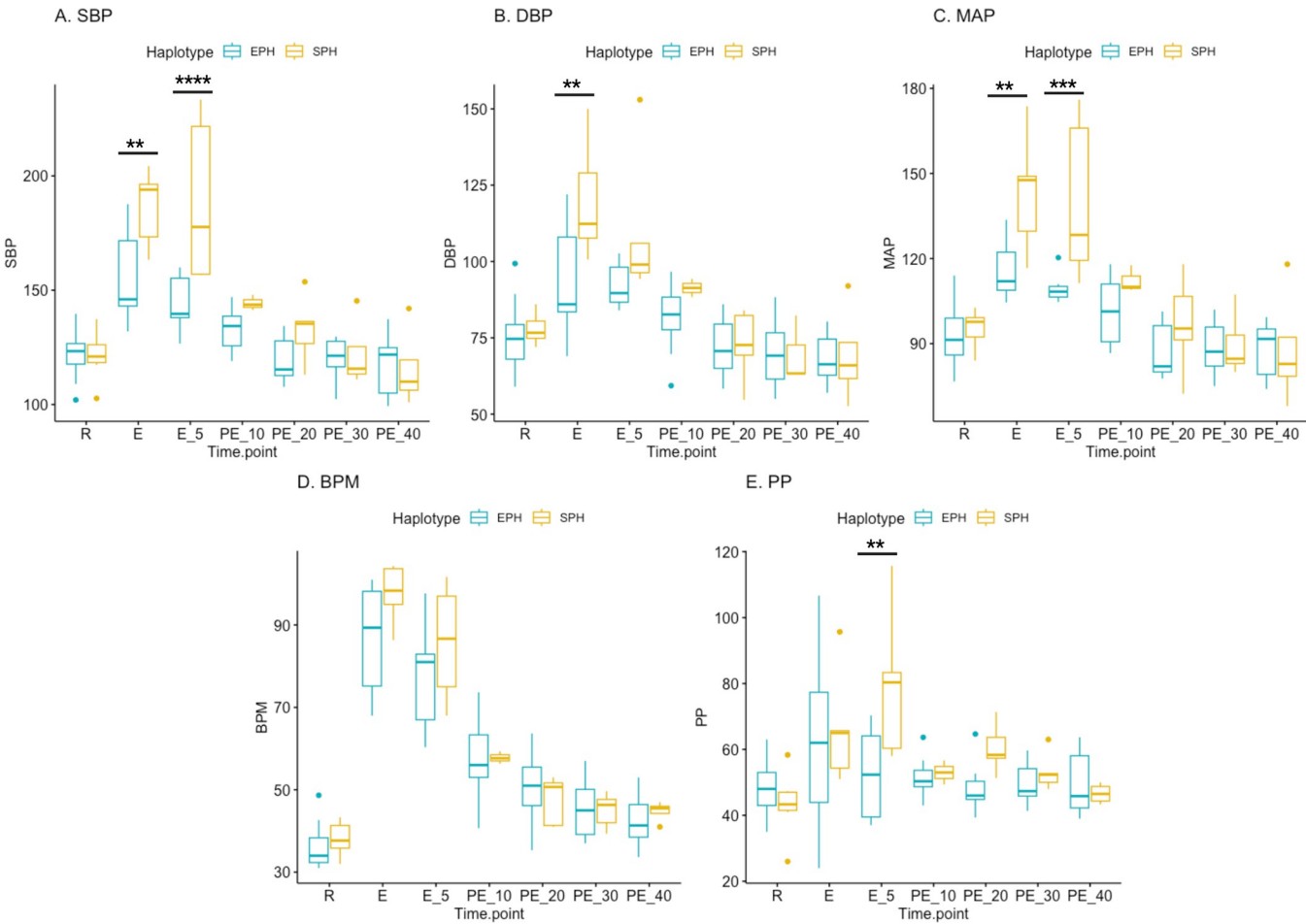

**Fig 4. Blood pressure measurements during and after exercise in horses homozygous for the sub-elite performing haplotype (SPH, n = 5–7) and horses homozygous for the elite-performing haplotype (EPH, n = 7–17).** A. SBP: Systolic blood pressure, B. DBP: Diastolic blood pressure, C. MAP: Mean arterial blood pressure, D. BPM: Beats per minute, E. PP: Pulse pressure. PP is defined as the systolic blood pressure minus the diastolic blood pressure. R = Rest, E = During exercise, directly after the uphill interval, E_05 = During exercise, 5 minutes after the uphill interval, PE = Post exercise.

performing multiple experiments, to first refine a selective sweep region associated with racing performance, followed by comparative transcription factor binding analyses and 3D genome interaction mapping, as well as horse plasma ELISA and blood pressure measurements, we identified a regulatory element comparatively active in human endothelial cells, potentially affecting horse blood pressure regulation and athletic performance. Here, we demonstrate a regulatory role of a potentially admixed genomic region, advantageous for harness racing performance in horses, and show that this region interacts with several well-known blood pressure linked genes, as identified in human GWAS studies [21,30].

The intergenic selective sweep identified in 2018 [15] is located 50 kb downstream of the *EDN3* gene. Given both the known role of EDN3 in blood pressure regulation and its proximal location, we hypothesized that 1) the identified non-coding region harbored a regulatory element that acted on the *EDN3* gene and 2) a key phenotype for elite athletic performance is blood pressure regulation. Our results demonstrated significant associations between the two minimal haplotypes identified in CBTs and SBs and multiple racing performance traits. In addition, in comparison with SPH homozygotes, horses homozygous for the EPH had significantly higher levels of EDN3 in their plasma and lower blood pressure readings during exercise.

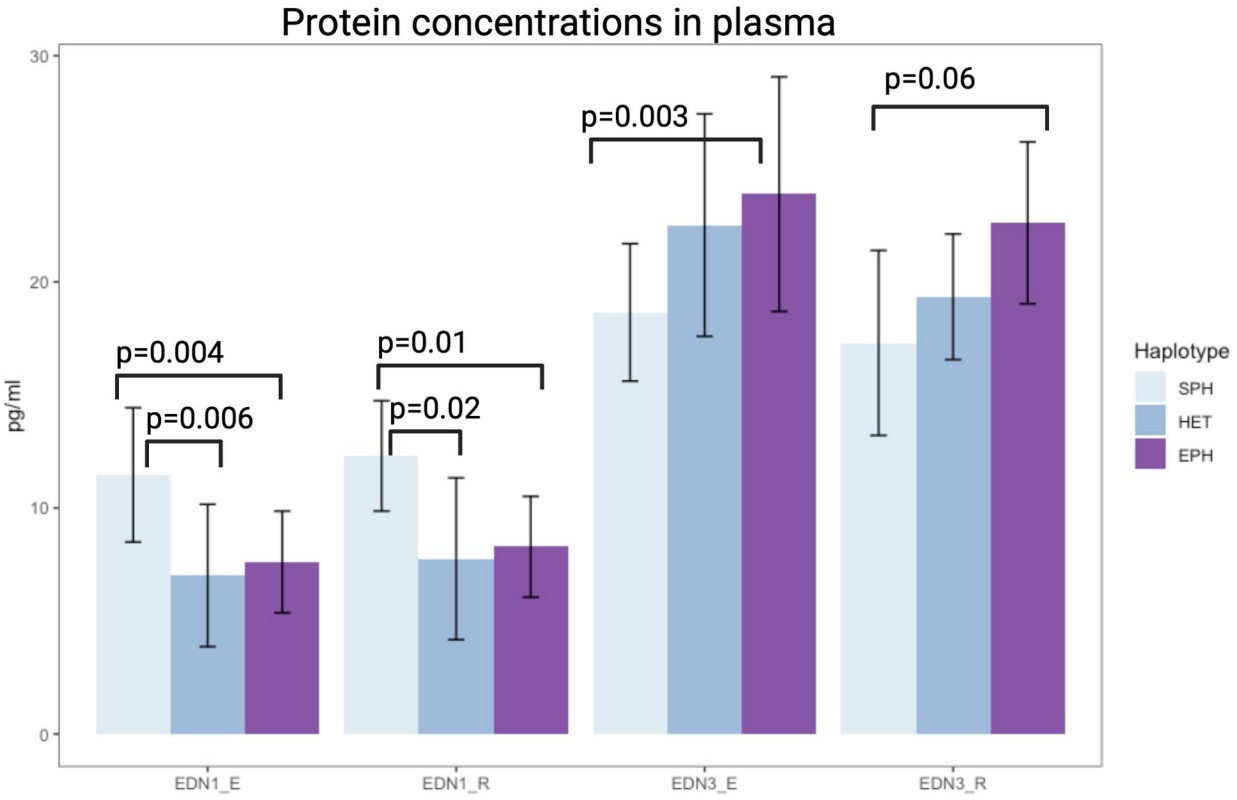

**Fig 5. Plasma concentrations (pg/ml) of EDN1 and EDN3 at rest (R) and during exercise (E) in horses homozygous for the sub-elite performing haplotype (SPH) (n = 8), horses heterozygous for the EPH and SPH (HET) (n = 11) and horses homozygous for the elite-performing haplotype (EPH) (n = 21).** ANOVA and Tukey´s HSD tests were performed. Standard deviations are presented as error bars.

The role of endothelins in blood pressure regulation is well studied. Endothelins are involved in one of the most potent vasoregulatory systems, where EDN1 and EDN3 have opposite effects and act synergistically to regulate blood pressure [43]. EDN1 increases blood pressure by vasoconstriction, while EDN3 is involved in nitric oxide release, in turn resulting in vasodilation and decreased blood pressure [43]. Further, EDN3 also stimulates the secretion of vasopressin, which increases blood volume by retaining water in the kidneys, while EDN1 has the opposite effect in the same organ [44–47]. Although EDN3 clearly contributes to blood pressure regulation, and was a strong candidate for our regulatory region, in the absence of appropriate horse samples, we used already produced HiCap datasets from 12 different human cell types, to investigate the comparative chromatin interaction profile of our identified region. Spanning 12 cell types covering multiple organs and life stages, including iPS cells and a neural cell line (S5 Table), the results showed that interactions were only observed between our putative regulatory region and different genes in human primary vascular endothelial cells. Endothelial cells make up the inner layer of blood vessels, and regulate the exchange between the blood and the surrounding tissues, by adjusting the vessel diameters according to the tissue demand [48]. Endothelial cells are critical regulators of the vascular tone and blood pressure regulation, as they influence the contractile ability of vascular smooth muscle cells within small arteries [49].

From the human HiCap data analysis, genes with direct interactions with our putative regulatory region included *GNAS*, *SPO11* and *ZNF831*, all with orthologues and conserved gene order in horses (Fig 1C). Also included was the human specific lincRNA *RP4-614C15.2* (Fig 2). Each of these genes encode proteins with known functional roles related to endothelial dysfunction but are certainly not limited to this function. *GNAS* is a complex locus which is imprinted in humans and mice [50,51]. It gives rise to multiple gene products through the use of both alternative promoters and splicing mechanisms [52]. This inclusion of varied first exons together provides instructions for the alpha subunit of the protein complex, called a guanine nucleotide-binding protein (G protein) [53]. This transmembrane protein complex is involved in calcium and potassium ion concentration changes within cells [54,55]. These changes are crucial for regulating heritable traits such as cardiac output and peripheral vascular resistance. In fact, blood pressure regulation in humans is a classical complex genetic trait with heritability estimates of 30–50% [56]. In line with this, GWAS of blood pressure measures has identified hundreds of genetic variants associated with systolic- and diastolic blood pressure as well as hypertension in humans [30]. A genomic region frequently identified in human GWAS for blood pressure measures is the wider *GNAS-EDN3* region [21,22]. The *SPO11* (Initiator Of Meiotic Double Stranded Breaks) gene is involved in DNA damage repair [57], and the stressors leading to DNA damage can be found during exercise.

In both humans and horses, intense exercise induces a physical stress response, in order to supply optimal energy and oxygen for the working muscles. Endothelial cells in the vascular wall have a significant role in maintaining blood flow by regulating the diameter of the blood vessels and by preventing the blood from clotting [58,59]. The blood flows more efficiently, and with less turbulence if the blood vessels are wide and if the blood is less viscous [60]. As such, blood vessels must be dilated during intense exercise, in order to supply working muscles with sufficient energy and oxygen. An increasing body of evidence suggests that oxidative stress, which results in an excessive generation of reactive oxygen species (ROS), is vital in the pathogenesis of hypertension [61]. Oxidative stress and inflammation significantly contribute to vascular remodeling by promoting exaggerated contractility and proliferation of vascular cells [62]. These factors also favor DNA damage, thereby linking SPO11 to endothelial dysfunction. It should be noted that several genes with known function in DNA damage and response pathways previously have been associated with pulmonary arterial hypertension [63]. *ZNF831* encodes a transcription factor that has been associated, via GWAS, with a wide range of human phenotypes, including body height [64], platelet component distribution width [65], hair color [66] and diastolic blood pressure [21,67]. The *ZNF831* mRNA is highly expressed in whole blood and spleen, and the protein is identified in the heart and T-lymphocytes [68], making it a likely candidate transcription factor for endothelial cells. While not orthologous in horse, the human lincRNA RP4-614C15.2 is a gene with a known role in angiogenesis [69]. LincRNAs are found explicitly in the nucleus, functioning in cell differentiation and identity. Imminent evidence suggests that long non-coding RNAs (lncRNAs) play critical roles in pulmonary vascular remodeling and pulmonary arterial hypertension (PAH) [70]. LncRNAs are implicated in regulating chromatin structure, thereby leading to pulmonary arterial endothelial dysfunction by modulating endothelial cell proliferation, angiogenesis, endothelial mesenchymal transition, and metabolism [71]. Targeting epigenetic regulators may lead to new, potential therapeutic possibilities in treating PAH [71]. It is clear that the search for potential lncRNAs regulating this wider genomic region in horses are required, but even so, these four directly interactive genes comprise promising candidates that warrant further investigation. The results observed from the human HiCap data were supported by the analysis of ChIP-seq data provided by the equine section of FAANG [72,73]. We demonstrated that there is an active enhancer in lamina, located in our 5.5 kb region (Fig 1). Laminae can be described as

finger-like protrusions of tissue in the hoof and is affected in equine laminitis. There is plentiful evidence that a compromised blood flow through the lamina develops in horses with laminitis e.g., [74], but little is known about the cause. One piece to this puzzle has shown that the concentration of EDN1 in laminar connective tissues obtained from laminitic horses was higher than in non-laminitic horses [75,76]. Horses with laminitis are hypertensive [77], consistent with digital hypoperfusion. All in all, these findings may be linked to the active enhancer in our 5.5 kb region in lamina.

Each SNV within the minimum shared 5.5 kb span was found to segregate in the genome of horses that lived before the rise and spread of the modern domestic lineage, around ~4,200 years ago (DOM2) [40]. Temporal allelic trajectories portray an increase in the alternative allele frequency, from 7,500–5,500 years ago, prior to the earliest archaeological evidence of horse husbandry [39,78]. This concerted rise in alternate allele frequency following the spread of the DOM2 lineage is compatible with a haplotype undergoing positive selection, possibly following artificial selection for horses showing improved athletic performance. Interestingly, the allele trajectories are in striking contrast with those previously described at the *myostatin* locus, which is driving performance in short-distance racing [8]. In that case selection was recent, indicated to start only within the last 1,000 years. It may be that performance traits represent recurrent selection targets over the history of horse domestication.

Although the current study demonstrated statistically significant differences in horse EDN1 and EDN3 levels between the different haplotype groups, there were no observed differences in blood pressure at rest. In adult horses, normal resting blood pressure is approximately 130/95 mmHg (systolic/diastolic) [79], in line with the values observed in the current study. In humans, it has been demonstrated that elevated blood pressure at rest is negatively correlated with athletic performance [80]. Individuals with elevated blood pressure had significantly lower maximal oxygen consumption, ventilatory anaerobic thresholds, and heart rate (HR) reserves (difference between an individual's resting HR and maximum HR) [80]. However, unlike humans, high blood pressure in horses is very uncommon, and most blood pressure measurements on horses are performed to evaluate and monitor hypotension. An increase in blood pressure at rest is most commonly seen as a result of diseases such as laminitis, chronic renal failure, or equine metabolic syndrome [81–83].

Perhaps the differences in observed blood pressure measurements between humans and horses is driven by species differences in the red blood cell reservoir, the spleen [60]. A previous study on horses demonstrated a significant correlation between blood pressure and spleen volume, with the splenic volume decreasing when hypertension was induced [84]. The equine spleen reservoir is larger than in any other domestic animal, and when there is increased demand, erythrocytes stored in the spleen can be released into the system [60]. Given the relationship between spleen size and blood pressure, further studies examining the relationship between the identified regulatory region, spleen size, and blood pressure represent promising avenues for enhancing the understanding of the cardiovascular system in horses in relation to exercise. While it may be viewed as a limitation, in the current study, blood pressure was measured using an external cuff. In general, direct blood pressure measurements via a fluid-filled cannula inserted into an artery are more accurate. However, the indirect blood pressure measurements are less invasive and horse cuff protocols have previously been successfully evaluated [85].

The regulatory potential of the region identified in this study is likely not limited to blood pressure regulation. For example, the EDN3 ligand has a broad expression across a range of tissues and has been implicated in diseases such as Hirschsprung's disease, Multiple Sclerosis, and Waardenburg syndrome in humans, as well as melanocyte development in mice [86–90]. Also, the *GNAS* gene encodes a G protein with multifaceted functionality. Much of vertebrate

physiology is based on the versatile functionality of G protein complexes associated with their G protein coupled receptors (GPCR) [91]. Notably, the G protein encoded by the *GNAS* gene helps stimulate the activity of an enzyme called adenylyl cyclase [92]. Adenylyl cyclase is the key enzyme that synthesizes cAMP and it is involved in cellular processes that help regulate the activity of endocrine glands such as the thyroid, pituitary gland, gonads, and adrenal glands [93,94]. In addition, adenylyl cyclase is thought to influence bone development signaling pathways, limiting the spatial production of bone with tissue [95]. Further, we see a trend associated with the haplotype and conformational type differences across horse breeds, warranting further studies. Another intriguing avenue is via thyroid hormone regulation, a network with profound effects on the cardiovascular system [96,97]. Therefore, it is unsurprising that the 15 protein-coding genes that directly, or indirectly, interacted with the putative regulatory region, showed gene ontology terms related to thyroid hormone regulation in an enrichment analysis (S7 Table). Thyroid hormone likely mediates the new demands and adaptations on the cardiovascular system needed for elite performance.

Taken together, this study underscores the pivotal role of a specific genomic region in regulating endothelial tissue integrity across species boundaries. By shedding light on the intricate mechanisms governing blood pressure modulation, these findings hold promise for advancing our understanding of cardiovascular health and disease. Moreover, the implications extend beyond blood pressure regulation, potentially encompassing a myriad of physiological processes influenced by hormone signaling. This elucidation of molecular interactions underscores the interconnectedness of mammalian physiology and highlights avenues for future research into therapeutic interventions and preventive measures against cardiovascular disorders.

## Conclusion

By combining state of the art genomics with the careful phenotyping of equine athletic performance and blood pressure measurements, we have identified a 5.5 kb potential regulatory region distal to *EDN3*, but with influence on multiple genes. The application of comparative interactome and bioinformatics datasets, allowed us to suggest that this is an enhancer region regulating the transcription of multiple blood pressure relevant genes in endothelial cells. These results are key to understanding the biological mechanisms behind blood pressure regulation in both human and horse elite athletic performance.

## Materials and methods

### Ethics statement

Horse blood sample collection was approved by the ethics committee for animal experiments in Uppsala, Sweden (number: 5.8.18-15453/2017 and 5.8.18-01654/2020). Human samples were collected following approvals for the Human Research Ethics Committee at Karolinska Institute (application number 2006/784-31/1 and 2012/1633-31/4), Solna, Sweden. Written informed consent was obtained from all the individuals according to the Declaration of Helsinki and methods were carried out following relevant guidelines.

### Horse material and DNA extraction

A summary of the various horses used across the experiments detailed below, is presented in S2 Table. Metadata and genotype information for all tested horses is included in S12 Table. Genomic DNA was extracted from hair roots or blood samples. For hair preparation, 186 μL of 5% Chelex 100 Resin (Bio-Rad Laboratories, Hercules, CA, USA) and 14 μL of proteinase K (20 mg/mL; Merck KgaA, Darmstadt, Germany) were added to each sample. The mixture was

incubated at 56˚C for two hours at 600 rpm and inactivated with proteinase K for 10 mins at 95˚C. Blood samples were extracted on the Qiasymphony instrument with the Qiasymphony DSP DNA mini or midi kit (Qiagen, Hilden, Germany).

## Racing performance association analysis with known selective sweep

All horse genomic coordinates in the study refer to the EquCab3.0 assembly [98]. In 2018, a 19.6 kb selective sweep (ECA22:46,702,297–46,721,892; EquCab3.0) was identified in CBTs [15]. Subsequently, the racing results and Illumina SNP670K Genotyping BeadChip genotypes from 378 CBTs were used to identify five SNVs ($r^2$>0.9 sweep region) as significantly associated with elite racing performance in that breed [15]. In this study, we repeat this sweep-performance analysis using horse metadata and 670K Axiom Equine Genotyping Array genotypes available from 629 CBTs [99]. The 629 CBTs include the 378 horses used in the 2018 study [15]. To define the region for analysis from within the sweep, we centered our analysis on the most significant variant from the 2018 study: rs69244086-T, and used available 670 K Axiom Equine Genotyping Array genotypes to calculate pairwise $r^2$ across the original 19.6 kb region. Variants with $r^2$>0.6 were included in the association analysis. Given the high LD, tests were not considered independent, and so the significance threshold was set at P ≤ 0.05. The available metadata included a) pedigree information, b) *DMRT3* gait associated genotypes [12] and c) performance data as individual race records for each horse (Swedish Trotting Association (Svensk Travsport) and the Norwegian Trotting Association (Norsk Rikstoto)). The *DMRT3* variant rs1150690013 (ECA23:22391254) was genotyped using StepOnePlus Real-Time PCR System (Thermo Fisher Scientific, Waltham, MA, USA) with the custom TaqMan SNP genotyping assay [100]. Reactions (15 μl) contained 1.5 μl DNA, 0.38 μl Genotyping Assay 40X, 7.5 μl Genotyping Master Mix 2X, and 5.62 μl deionized water. The *DMRT3* variant was included as a factor in the association model, as previous studies have demonstrated major effects of the variant on harness racing performance [12,14]. Only performance data from competitive races were included in the analysis (i.e., premie and qualification races were excluded). The performance data included: number of starts, number of wins, number of placings (the number of times a horse finished a race in first, second or third place), fastest kilometer (km) time in a race where the horse did not gallop (in seconds), earnings and earnings per start. The earnings for most horses were recorded in Swedish currency (SEK). Earned prize money in Norwegian currency was converted to SEK based on the average exchange rate for the specific race year.

All statistical analyses were performed in the R statistical environment versions 4.1.1 and 4.2.2 [23]. The performance data were tested for normality by computing the skewness coefficient using the package moments v0.14. Non-normally distributed values were transformed, i.e., log transformed values (log10 +1) were used for wins and placings, number of starts were square root transformed and earnings and earnings per start were transformed according to the previously reported formula (ln(earnings + 1000)) [101]. Genetic variants were tested in a linear model using ANOVA as a *post hoc* test. Number of starts, age at first start, sex, birth year, country of birth, and the *DMRT3* genotypes were included as fixed effects, when significant. Significant values (P ≤ 0.05) were further tested using the function lsmeans (Least-Squares Means) with the package emmeans followed by the multiple comparison test Tukey's HSD-test.

## SNV, indel and SV variant discovery

To find variants not included on the 670K Axiom Equine Genotyping Array, but with the potential to influence racing performance we performed Illumina short-read WGS. Two CBTs and two SBs were selected based on their rs69244086 genotype and their performance

earnings. For each breed, one horse with high earnings per start and homozygous T at SNV rs69244086, and one horse with low earnings per start homozygous C at SNV rs69244086, were selected for sequencing (S2 Table). Paired-end 150 bp Illumina HiSeqX data was generated to a depth of 15X. Read data were processed according to the GATK 3.8–0 best practices (https://software.broadinstitute.org). Briefly, raw data were trimmed with TrimGalore 0.4.4 and quality checked with QualiMap 2.2. The reads were aligned to the EquCab3.0 reference genome (98) using BWA-MEM 0.7.17. SAM files were converted to BAM files using SAMtools 1.8, and Picard 2.10.3 was used to sort the BAM files and remove potential PCR duplications. The BAM files were recalibrated (BaseRecalibrator and PrintReads) and small variants were called with HaplotypeCaller using a list of known variants for calibrations (NCBI release 103, https://www.ncbi.nlm.nih.gov). Structural variants were called using default parameters in FindSV v0.4.0 combining the programs Manta, TIDDIT and CNVNator (https://github.com/J35P312/FindSV).

## Variant filtration and genotyping

In total, 84 variants, including 78 SNVs and 6 indels of various sizes, were discovered from the WGS data. From the SVs, only a 400 bp deletion (ECA22:46,714,602–46,715,003) matched the required segregation pattern of present in CBTs and SBs, reflective of a selection on crossbreed admixture [15]. The 400 bp deletion was subsequently genotyped in a further 497 CBTs with performance data and available blood sample. The genotyping was performed with digital droplet PCR (ddPCR) using the ddPCR Supermix for Probes (No dUTP) (Bio-rad) protocol. An amplicon of 70 bp (primer sequences; fwd: CTGAGCACAGGGCAGTGT, rev: GAGCGGACAAGAGCGATTG and probe FAM–CCGGGAAAACAGCCCCTTCC) was used to detect the deletion. A 97 bp amplicon of the Glyceraldehyde-3-Phosphate Dehydrogenase (GAPDH) gene was used as a control (fwd CGATGCTGGTGCTGAATATGTT, rev: GGTCAACTCCCCTCATCTTTAGC and probe sequence TCTTCACTACCTTGGAGAAG with HEX fluorophore). Genomic DNA restriction enzyme digestion of DNA was performed in the ddPCR reaction using 1 ul of FastDigest TRU1I EA restriction enzyme (Thermo Fisher Scientific). Droplets were generated with Automated Droplet Generator (Bio-Rad Laboratories Inc). The PCR reaction volume was 20 ul and the PCR program included enzyme activation at 95˚C for 10 min, 40 cycles of denaturation at 94˚C for 30 seconds, and annealing at 60˚C for 1 minute, followed by enzyme deactivation at 98˚C for 10 minutes. The samples were run in the QX100 or QX200 Droplet reader, and the data were analyzed in the QuantaSoft Software (Bio-Rad Laboratories Inc). Thresholds were set manually in the 2D amplitude tab.

For the smaller variants, technical limitations restricted us to 25 multiplexed SNVs (including indels) (MassARRAY Assay Design Suite v2.2 software, Agena Bioscience, San Diego, CA, USA). As such, discovery variants were filtered for inclusion based on i) genotype segregation patterns from a previous pooled genotype experiment [16] and then ii) for even spacing across the 19.6 kb sweep region. The pooled experiment included the three Nordic horse breeds represented in the original 2018 selective sweep analysis, CBTs, SBs and the NSD [15]. Following the pattern of suspected introgression of SB into CBT, WGS discovery variants were retained if their genotyping pattern in the pooled data was similar between the CBT and SB pools, and discordant in the NSD pool. After filtering, 24 SNVs and one INDEL were selected for MassArray genotyping.

To assess breed or species specificity, the 25 multiplexed variants were genotyped across 394 samples (S13 Table). This included 210 CBTs, plus 181 samples from 11 other breeds, as well as 3 Przewalski horses. The representative samples were selected to include both traditional performance horse breeds (e.g., Thoroughbreds), and non-performing breeds (e.g.,

Shetland pony) (S13 Table). The genotyping was performed using iPLEX Gold chemistry and the MassARRAY mass spectrometry system [102] (Agena Bioscience) at the Mutation Analysis Facility at Karolinska University Hospital (Huddinge, Sweden), according to Agena Bioscience's recommendations. In brief, analytes were spotted onto a 384-element SpectroCHIP II array (Agena Bioscience), using Nanodispenser RS1000 and subsequently analyzed by MALDI-TOF on a MassARRAY Analyzer 4 mass spectrometer (Agena Bioscience). Genotype calls were manually curated based on concordance calls from two operators using MassARRAY TYPER v4.0 Software (Agena Bioscience). Due to design limitations, the known performance associated SNV rs69244086 [15] was not included as part of the MassArray. Rather a custom TaqMan SNV assay (100) was designed, and assayed using the StepOnePlus Real-Time PCR System (Thermo Fisher Scientific). Here, due to DNA availability, a reduced set of 242 samples was genotyped (183 CBTs, 38 SBs, 5 NSD, 2 Ardennes, 4 Finnhorses, 1 Fjord horse, 2 Gotlandsruss, 2 Icelandic horses, 2 shetland ponies and 3 Pzrewalski).

### Association analyses with discovery variants

For the SV association analysis, a linear model was developed as per above section, "Racing performance association analysis with known selective sweep", including horse metadata and the genotypes from the 400 bp deletion for 497 CBTs and 41 SBs. A haplotype block for association analysis was then defined as all SNVs from the MassArray that were in pairwise LD $r^2 >$ 0.95 with rs69244086 (LD-function in R). Prior to association analysis, maximum likelihood estimates were used to determine the 14 SNVs haplotypes with the haplo.em function in the haplo.stat package [103]. In total, 166 CBTs and 38 SBs were available for analysis. Haplotype racing performance association tests used a GLM regression analysis using the haplo.glm function from the haplo.stats package in R [103]. The model included the effects of sex, age, country of registration, *DMRT3* genotype (only for the CBTs) and number of starts, when significant. All CBTs were genotyped for the *DMRT3* mutation, using a custom TaqMan SNP genotyping assay [100] as described in the section "Racing performance association analysis with known selective sweep". CBTs not typed for the *DMRT3* genotype were excluded from the analysis.

Haplotypes with a frequency < 2% were considered rare, and therefore not analyzed for association with performance. A minimum haplotype span was based on haplotype sharing between CBTs and SBs, and was used to define the elite-performing haplotype (EPH) and the sub-elite performing haplotype (SPH). As the core of the EPH and SPH haplotypes are defined by extremely high pairwise LD ($r^2 > 0.95$) alleles, any of the variants within this span can be used as proxy for the haplotypes themselves. We therefore extended the number of horses we could include in the association analysis by genotyping 519 CBTs and 271 SBs for rs69244089 using a custom TaqMan SNP genotyping assay [100], and methods described in the section "Racing performance association analysis with known selective sweep". Three CBTs lacked *DMRT3* genotypes, and were excluded from the analysis. In total, 516 CBTs born between 1994 and 2017, and 271 SBs born between 2005 and 2014 were then available for association with racing performance. A linear model was performed in R. Sex, age, birth country, number of starts, and *DMRT3* genotype were used as fixed effects. Least square means and Tukey were used as post hoc tests.

### "Performance" SNV frequency in other populations

As per the rs69244086 SNV in section, "Variant filtration and genotyping", horses from 13 different breeds as well as Pzrewalski horses, were also genotyped using the TaqMan assay for rs69244089, in order to investigate the distribution of this variant. The Hardy-Weinberg

equilibrium test (hwe) from the gap package in R, was used to test if the genotypes observed deviated from HWE in SB and CBT.

## Comparative genomics to infer haplotype function

In the absence of empirical data from genotyped horses, we used comparative data from horses and human sources. For horses, ChIP-seq data were provided by the equine section of FAANG [72,73]. These samples from project PRJEB35307 were generated from two Thoroughbreds and included four histone marks (H3K4me1, H3K4me3, H3K27ac and H3K27me) across nine tissues (adipose, brain, heart, lamina, liver, lung, muscle, ovary and skin). After local download, data integrity was confirmed with checksums. Bam files were sorted with samtools and bedgraphs were created with bedtools [104,105]. Bigwig files were viewed as custom tracks in UCSC, EquCab3.0.

To assess the per SNV potential intersection from the minimum haplotype and surrounding genes, we had access to twelve different human HiCap [106] datasets produced at the Sahlén Laboratory (S5 Table). HiCap combines Hi-C and targeted sequencing, to obtain high-resolution genome interaction maps [19]. In short, live cells are crosslinked using formaldehyde, to immobilize chromatin. The cells are lysed and chromatin is digested using a frequent cutter restriction enzyme (such as 4-cutter *DpnII*). The digested chromatin is then end-repaired using biotinylated nucleotides, and ligated back to itself, to capture the proximity information (ends that are close to each other in 3D nuclear space are more likely to be ligated than those that are far away). Once the ligation is complete, the crosslinking is reversed using heat, proteins are removed and DNA is purified, completing the Hi-C step [107]. A sequencing library is then prepared using the Hi-C material. The Hi-C library is hybridized to a set of probes targeting around 21,000 promoters, to obtain only the proximities of promoters, providing up-to 110-fold increase in promoter interaction resolution [106]. Custom designed probes were purchased from Agilent Technologies Inc (Santa Barbara, CA, USA) including the reagents for sequencing library preparations. The Hi-C material was captured and sequencing libraries were prepared using HS2 XT Library Prep Kit (Agilent Technologies Inc, article number: 5190–9685, 5500–0146 and 5191–6686) with small modifications [106]. In several steps, the final libraries were sequenced, on average, ~640 million paired-end for each experiment, at the National Genome Infrastructure Sweden (paired end 2x150 bases). The reads were mapped to hg38 assembly using Bowtie2 (version 2.2.9) using the -very-sensitive option [108]. HiCaptools was used to call interactions in all samples using default settings. GENCODE (version 37) comprehensive gene annotation using only reference chromosomes [109] was used to annotate promoters. We required at least four supporting pairs and true P-value less than 0.05 for each interaction. We overlapped all promoter-interacting regions with the variants around the EDN3 gene using bedtools intersect function via the bedr package v1.0.7 [110] implemented in R (version 4.2.1) [23] with no padding. We searched the twelve HiCap datasets for the interactions of the minimal haplotype block lifted to either hg19 (chr20:57950706–57964856) or hg38 (chr20:59375651–59389801) depending on the dataset. We used interaction sets called at FDR 0.1 thresholds except the vascular endothelial cells [111]. No interactions were present in any of the cell types apart from the vascular endothelial cells. The primary vascular endothelial cells were collected from 16 patients undergoing elective open-heart surgery at the Cardiothoracic Surgery Unit, Karolinska University Hospital, Stockholm, Sweden. HiCap was performed on 2.5 million cells derived from each patient according to the latest protocol [111]. All interactions that were statistically significant in each patient (P < 0.05) were called. There were, in total, 49 interactions overlapping with variants. For comparative plotting on the horse reference genome, interactions that were present in at

least two individuals were considered. A network generation figure was generated using the Flourish visualization app (https://flourish.studio). To reduce complexity of the network figure, only interactions that were present in at least five individuals were included. For all genes with significant interactions with our region of interest, we performed a gene enrichment analysis using the ToppGene Suite [112].

We could not further fine map the minimum haplotype to identify candidate causative SNVs in the region. However, the impact on transcription factor binding was assessed with MotifbreakR [34] in R v4.2.2 and for 14 SNVs in the minimum shared 5.5 kb locus. Using forgeBSgenomeDataPkg (included in motifbreakR) and the UCSC equCab3.2bit file (http://hgdownload.soe.ucsc.edu/goldenPath/equCab3/bigZips/), we created the genome library required for motifbreakR as it is not supplied with this package. We assessed the impact of allele difference between the EPH and SPH haplotypes with hocomoco motifs [33], default position weight matrix to score each, at threshold of $1 \times 10^{-4}$, before retaining the strong category of effects.

## Ancient DNA screening

We leveraged the availability of an extensive ancient genome time-series in the horse to assess the temporal trajectory and spatial distribution of the SNVs in the minimum shared haplotype within the CBTs, in the past. More specifically, we re-aligned against the EquCab3.0 reference genome (98) supplemented with the Y-chromosomal contigs from [113] and a sub-selection of the sequencing data from [4,36,38,39,114,115], representing a total of 431 ancient genomes. Alignment files were generated using the Paleomix pipeline [116], and further rescaled and trimmed, following the procedures described in [40]. This procedure ensured minimal sequencing error rates, especially at sites potentially affected by post-mortem DNA damage. The number of occurrences of each individual allele was counted for each individual using ANGSD (v0.933), with the -baq 0, -remove_bads -minMapQ 25 -minQ 30 -rmTriallelic 1e-4 -SNP_pval 1 -C 50 parameters [117]. The resulting occurrences were tabulated together with metadata available for each individual, consisting of their corresponding archaeological site (name and GPS coordinates) and their average calibrated radiocarbon date (or age as assessed from the archaeological context otherwise). The tabulated file was visualized using the map-DATAge package [35], which provides spatial distributions within user-defined time periods and/or spatial ranges and estimates allelic trajectories (i.e., allelic frequencies through time) based on the resampling of individual allelic counts. The final allelic trajectories were assessed considering time bins of 2,000 years (step-size = 500). To explore the genetic variation for the SNVs within the haplotype, we considered the 14 polymorphic sites i.e., rs395117226, rs397265747, rs396474304, rs69244081, rs69244084, rs69244085, rs69244086, rs69244088, rs69244089, rs396281591, rs394573286, rs69244091, rs69244093 and rs69244095.

## Exercise test, sample collection and blood pressure measurements

**Horse material.** Thirty horses were enrolled in the exercise test (S2 Table). All horses were genotyped for SNVs rs69244086 and rs69244089 using the StepOnePlus Real-Time PCR System (Thermo Fisher Scientific) with the custom TaqMan SNP genotyping assay as above [100]. The two SNVs were used as a proxy for the haplotype variants homozygote EPH (TT respectively CC) or homozygote SPH (CC respectively TT). Horses were then divided into three groups based on haplotype variant, i.e., homozygote EPH or SPH and HET. The horses were carefully selected to include an even sex and age distribution across the different groups. The age of the horses ranged from two to 13 years (mean age = 5.5 years). All horses, except for the two-year old, were in competing condition. However, due to various reasons including

practical problems, blood pressure measurements were not taken from all horses at all time points and some horses only had measurements at rest (S2 Table). For the ELISA, plasma samples were collected from 40 horses, before and/or during exercise. As for blood pressure measurement, some horses only had samples collected at rest.

**Exercise.** All horses, except for the two two-year old horses, were challenged on the same standardized exercise session (n = 28). The test was performed on several days during autumn and winter of the years 2020 and 2021. Each horse, with a trainer, completed a 1.5-hour exercise route that included both flat and uphill interval training. This route was commonly used in their daily training program. A Polar M460 sensor with GPS (Polar Electro, Kempele, Finland) was used to record HR, speed, distance and amplitude during exercise. The horses performed the exercise on a track with hilly terrain in the forest. The uphill interval section consisted of the horses trotting four times up a hill with four degrees inclination. Each uphill interval lasted for five minutes with a distance of 1.7 km. The HR for each horse reached above 200 beats/minute. After the uphill intervals, the horses walked down to the flat part of the track, where they completed 10 minutes of intervals, at a speed of 35 km/h and HR of about 180–200 beats/minute. Finally, the horses jogged one km back to the stable at a speed of 15–20 km/h. During the exercise, the average HR reached above 190 beats/minute for approximately eight mins. Due to their young age the two-year old horses (n = 2) performed a standardized exercise session at the racetrack. Following warm-up, they performed a number of intervals with HR reaching over 200 beats/minute. The two-year old horses performed the exercise test at the same time.

**Blood pressure measurements.** The following measurements were collected with a Cardell Veterinary Monitor 9402 cuff (Cardell 10), at the middle coccygeal artery: systolic, diastolic, mean arterial pressure, pulse pressure, and HR. The same cuff size was used for all the horses. The monitor has previously been successfully evaluated for use in horses [85]. Blood pressure was measured at rest in the stable before the exercise and during exercise (directly and five minutes after the last uphill interval) (Fig 6). Additionally, blood pressure was

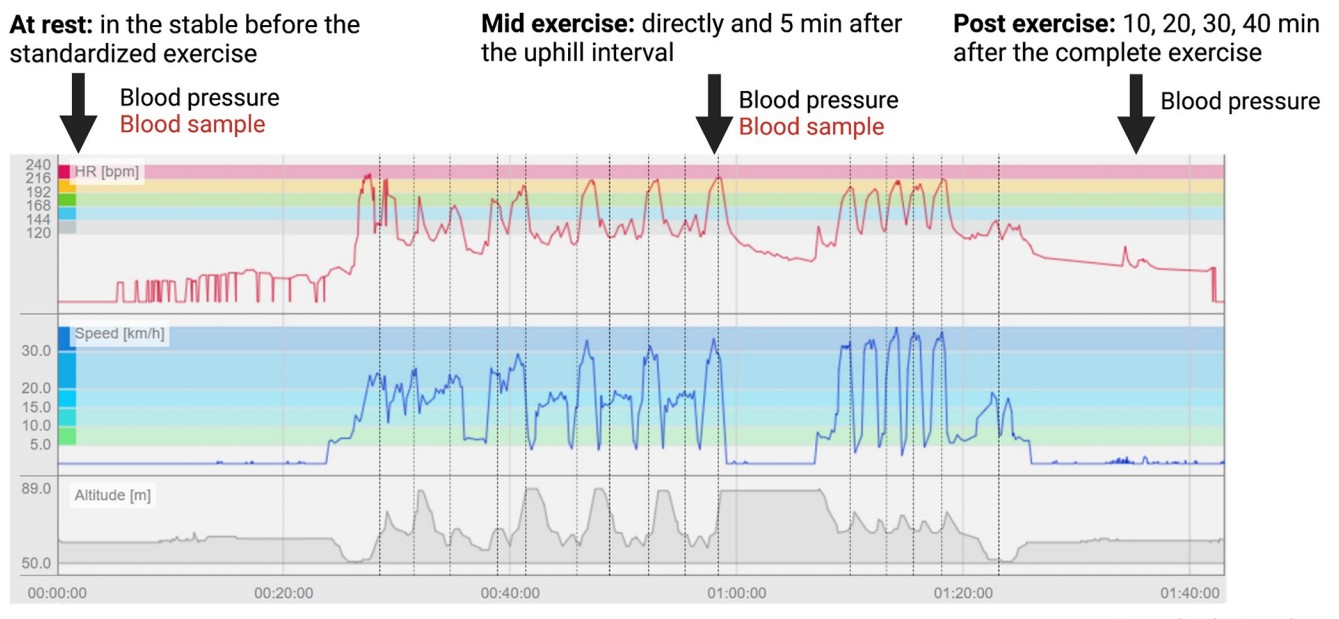

**Fig 6. A schematic overview of the parameters recorded during the exercise for each horse, including heart rate, speed and altitude.** The arrows indicate when blood samples and blood pressure measurements were taken.

measured every ten minutes after exercise, up to 40 minutes after the exercise was completed. In order for the horse to acclimate to the equipment, the Polar M460 sensor and the cuff were put on at rest, before the exercise route, and the horse was left in the box for 10 minutes before measurements were taken. Similarly, blood pressure was measured five times at rest for the horse to get used to the measurement procedure. The measurements with the highest and lowest systolic blood pressure values were removed. During and after exercise the blood pressure was measured three times and an average value was calculated. For the association analyses with blood pressure, the mean values of the triplicates were analyzed using Linear models and ANOVA, with pairwise tests using emmeans, corrected for multiple testing with Bonferroni. We also analyzed the correlation between the different blood pressure measurements and plasma EDN1 and EDN3 with ggplot and Pearson Correlation coefficient in R.

**Biological sample collection and analysis.** Blood samples were collected at rest before the exercise and during exercise directly after the last uphill interval. Blood was also collected for plasma extraction. Here, a sample was collected in an EDTA tube and mixed gently four times after sampling, to allow the EDTA to mix with the blood, then centrifuged at 1000 x g (3000 rpm) for 20 minutes as soon as possible and within two hours after sampling. The tubes were kept cold until centrifugation. The EDTA plasma was aliquoted in Eppendorf tubes and immediately frozen at -20˚C and within five days put at -80˚C until analysis.

**ELISA analysis of EDN1 and EDN3.** Plasma EDN1 concentrations were measured using the Horse Endothelin ELISA Kit PicoKine EK0945-EQ (Boster Bio, Pleasanton, CA, USA). Plasma EDN3 was quantified with the horse EDN3 ELISA Kit MBS9901200 (MyBioSource, San Diego, CA, USA). The analyses were performed according to the manufacturer's manual. A standard curve was created, and the samples were run in duplicates. Mean values of the concentration (pg/ml) were used in the statistical analysis, and the standard deviation (SD) and CV were calculated. Differences in concentrations between the three groups (EPH, HET and SPH) before and during exercise were analyzed using ANOVA and Tukey´s HSD test. Significance was set at $P \leq 0.05$.

## Supporting information

**S1 Table. Performance analysis results for the five significant SNPs in Coldblooded trotters.**
(XLSX)

**S2 Table. Information about all horses sampled and used for the various analyses.**
(XLSX)

**S3 Table. Location of discovery and genotyped variants with the minimum 5.5 kb region.**
(XLSX)

**S4 Table. Performance results for the 400 bp deletion in 497 Coldblooded trotters.**
(XLSX)

**S5 Table. The HiCap interactions of the putative regulatory element in different human cell types.**
(XLSX)

**S6 Table. The protein-coding genes that directly or indirectly interact with the regulatory region.**
(DOCX)

**S7 Table. Enrichment of 15 protein-coding genes directly or indirectly interacting with the putative regulatory element.**
(XLSX)

**S8 Table. List of all analyzed variable sites in the 5.5 kb region.**
(XLSX)

**S9 Table. Mean values ± SD for all blood pressure measurements.**
(DOCX)

**S10 Table. Plasma concentration of Endothelin 1 (EDN1) (mean + standard deviation) at rest and during exercise.**
(DOCX)

**S11 Table. Plasma concentration of Endothelin 3 (EDN3) (mean + standard deviation) at rest and during exercise.**
(DOCX)

**S12 Table. Metadata and genotypes for horses included in the study.**
(XLSX)

**S13 Table. List of horses included in the MassArray genotyping analysis.**
(DOCX)

**S1 Fig. Illustration of the five most significant high impact transcription factor binding changes that may result from a switch of alleles contained within the reference SPH to the non-reference EPH.** SNVs rs69244086 C>T (A) and rs69244089 T>C (G/A) (B) are illustrated.
(TIF)

## Acknowledgments

Thanks to all horse owners and trainers who participated in the study, The Swedish Trotting Association and the Norwegian Trotting Association for providing pedigree and performance data, Tytti Vanhala at the Department of Animal Breeding and Genetics, SLU, for help with genotyping, Anna Johansson at the National Bioinformatics Infrastructure Sweden at SciLife-Lab for bioinformatics advice, and the Department of Clinical Sciences Clinical training center (KTC) at SLU for providing the Cardell Veterinary monitor for the blood pressure measurements. We also thank Mats Pettersson for statistical advice and Anna Svensson´s laboratory at the Department of Clinical Sciences SLU for running ELISA. WGS was performed by the SNP&SEQ Technology Platform, Uppsala University. The facility is part of the National Genomics Infrastructure (NGI) Sweden and Science for Life Laboratory. The SNP&SEQ Platform is also supported by the Swedish Research Council and the Knut and Alice Wallenberg Foundation. The computations and data handling were enabled by resources provided by the National Academic Infrastructure for Supercomputing in Sweden (NAISS) and the Swedish National Infrastructure for Computing (SNIC) in Uppsala partially funded by the Swedish Research Council through grant agreements no. 2022–06725 and no. 2018–05973.

## Author Contributions

**Conceptualization:** Göran Andersson, Gabriella Lindgren.

**Data curation:** Maria K. Rosengren, Rakan Naboulsi, Ludovic Orlando, Magnus Åbrink, Pelin Sahlén, Jennifer R. S. Meadows.

**Formal analysis:** Kim Fegraeus, Maria K. Rosengren, Karin Lång, Artemy Zhigulev, Hanna M. Björck, Anders Franco-Cereceda, Per Eriksson, Pelin Sahlén, Jennifer R. S. Meadows.

**Funding acquisition:** Amanda Raine, Gabriella Lindgren.

**Investigation:** Kim Fegraeus, Maria K. Rosengren, Ahmad Jouni, Jennifer R. S. Meadows.

**Project administration:** Gabriella Lindgren.

**Resources:** Maria K. Rosengren, Ludovic Orlando, Karin Lång, Artemy Zhigulev, Hanna M. Björck, Anders Franco-Cereceda, Per Eriksson, Pelin Sahlén.

**Software:** Ludovic Orlando.

**Supervision:** Rakan Naboulsi, Brandon D. Velie, Amanda Raine, Beate Egner, C Mikael Mattsson, Göran Andersson, Jennifer R. S. Meadows, Gabriella Lindgren.

**Validation:** Maria K. Rosengren.

**Visualization:** Kim Fegraeus, Pelin Sahlén, Jennifer R. S. Meadows.

**Writing – original draft:** Kim Fegraeus, Jennifer R. S. Meadows, Gabriella Lindgren.

**Writing – review & editing:** Kim Fegraeus, Maria K. Rosengren, Rakan Naboulsi, Ludovic Orlando, Magnus Åbrink, Ahmad Jouni, Brandon D. Velie, Amanda Raine, Beate Egner, C Mikael Mattsson, Karin Lång, Artemy Zhigulev, Hanna M. Björck, Anders Franco-Cereceda, Per Eriksson, Göran Andersson, Pelin Sahlén, Jennifer R. S. Meadows, Gabriella Lindgren.

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
