## [Decision Letter · Decision Letter 0]

20 Sep 2023

Dear Dr Fegraeus,

Thank you very much for submitting your Research Article entitled 'An endothelial regulatory module links blood pressure regulation with elite athletic performance' to PLOS Genetics.

The manuscript was fully evaluated at the editorial level and by independent peer reviewers. The reviewers appreciated the attention to an important problem, but raised some substantial concerns about the current manuscript. Based on the reviews, we will not be able to accept this version of the manuscript, but we would be willing to review a much-revised version. We cannot, of course, promise publication at that time.

If you decide to revise the manuscript for further consideration at PLOS Genetics, please aim to resubmit within the next 60 days, unless it will take extra time to address the concerns of the reviewers, in which case we would appreciate an expected resubmission date by email to plosgenetics@plos.org.

We are sorry that we cannot be more positive about your manuscript at this stage. Please do not hesitate to contact us if you have any concerns or questions.

Yours sincerely,

Carrie J Finno, DVM, PhD

Guest Editor

PLOS Genetics

Gregory Barsh

Editor-in-Chief

PLOS Genetics

Reviewer's Responses to Questions

**Comments to the Authors:**

Reviewer #1: Fegraeus et al. report a follow-up study on earlier work regarding the genetics of athletic performance in horses. Starting from a previously identified and published sweep region of 19.6 kb downstream of the EDN3 gene, the authors were able to refine the associated region to 5.5 kb. This region harbors 14 single nucleotide variants (SNVs) that are in strong linkage equilibrium with each other. The authors determined that only two major haplotypes for these 14 SNVs segregate in the general horse population, which they termed elite performing haplotype (EPH) and subelite performing haplotype (SPH). Extrapolating from existing human data, Fegraeus et al. identify potential regulatory networks that provide new plausible and testable hypotheses how the variants might ultimately affect endothelin signalling and blood pressure regulation. The proposed phenotypic effect on endothelin serum levels and blood pressure was validated in horses with the different genotypes. Overall, this is a comprehensive and well presented study that enables a significantly improved understanding of the complex traits under investigation.

Specific comments:

(1)

The data availability statement is inacceptable. The raw data must be made publicly available prior to the publication of this manuscript and accession numbers must be given in the manuscript. Individual phenotypes and marker genotype data of the studied horses must be given in one or more supplementary tables to allow interested readers to repeat the presented analyses.

(2)

Figure 1: The figure would be much easier to grasp, if the displayed species were indicated in each of the four panels A-D (horse or human). The authors may want to consider to redesign panel D in order to better display the correspondence between horse and human. Alternatively, another supplementary figure may be added to display more details of the horse-human correspondence (e.g. a dot plot and/or a full sequence alignment).

(3)

Can you add some additional text with more information why the human sequence is twice as long as the horse sequence (11 kb vs 5.5 kb)? What is the repeat content in the two species in this region?

(4)

Figure S2: Please expand the figure legend and give more explanations. The legend should contain an easy to grasp list/explanation of all transcription factors, for which binding site consensus sequences are shown. Is a score or any other kind of measure available, how likely it is that these transcription factor will bind to the reference and alternate alleles? A supplementary table with all TF binding scores for all analyzed variable sites should be added.

(5)

Consider to replace SNP with SNV.

(6)

I assume that all horse genomic coordinates in this study refer to the EquCab3.0 assembly. If true, please state this explicitly at the beginning of the methods section.

(7)

Line 266: "Data not shown" is not allowed in PLoS Genetics publications. All underlying raw data must be made publicly available.

(8)

Line 389: Myostation -> myostatin

(9)

Line 664: Suit  Suite (?)

Reviewer #2: The authors describe the identification of a causative genomic region (5.5 kb/14 SNPs, regulatory element acting on EDN3 transcription) for blood pressure regulation and athletic performance in horses based on fine-mapping and comparative analyses. Then, the authors defined and identified an elite performing haplotype (EPH) and a sub-elite performing haplotype (SPH). EPH is associated with higher plasma levels of EDN3 and lower levels of EDN1, and lower exercise-related blood pressure.

This research is well organized, but the following revision or comment are necessary.

I have read this manuscript in a previous round of peer review. In the previous manuscript, it was unclear whether the 5.5 kb site truly functioned as an enhancer site, as there were no results from gene expression analysis. The new Human HiCap analysis performed in this study may have compensated for previous weaknesses.

However, this reviewer has some concerns about Human HiCap analysis. While the author's main research is conducted on horses, HiCap analysis is conducted on humans. Is it appropriate to conduct an analysis using cells from different species in this way? Has this approach been used in other studies?

So, the authors should discuss discrepancies with horse FAANG and human HiCap analyses.

The author conducted WGS to comprehensively identify variants, but the number of animals analyzed seems to be small. Is there a possibility that a causative variant exists at a site distant from the 5.5 kb site?

Reviewer #3: Please see attachment.

**Have all data underlying the figures and results presented in the manuscript been provided?**

Reviewer #1: **No: **Please see my comments ot the authors.

Reviewer #2: Yes

Reviewer #3: Yes

PLOS authors have the option to publish the peer review history of their article (what does this mean?). If published, this will include your full peer review and any attached files.

Reviewer #1: No

Reviewer #2: No

Reviewer #3: No

---

## [Decision Letter · Decision Letter 1]

5 Feb 2024

Dear Dr Fegraeus,

Thank you very much for submitting your Research Article entitled 'An endothelial regulatory module links blood pressure regulation with elite athletic performance' to PLOS Genetics.

The manuscript was fully evaluated at the editorial level and by independent peer reviewers. The reviewers appreciated the attention to an important topic but identified some concerns that we ask you address in a revised manuscript.

We therefore ask you to modify the manuscript according to the review recommendations. Your revisions should address the specific points made by each reviewer.

Yours sincerely,

Carrie J Finno, DVM, PhD

Guest Editor

PLOS Genetics

Gregory Barsh

Editor-in-Chief

PLOS Genetics

Reviewer's Responses to Questions

**Comments to the Authors:**

Reviewer #1: The authors have adressed my comments satisfactorily.

The data availability statement is now acceptable to me. I find it regrettable that the ethics permit for this study was issued by a Human Research Ethics Committee rather than a committee related to animal welfare (an IACUC). This study does not relate to human patients, it relates to animals. Furthermore, it is my understanding that this study is about variation of physiological blood pressure parameters, not a disease. I therefore think that the term "living patient" in the data availability statement is inappropriate. I do not understand why sequencing data of animals cannot be made publicly available. This sets a very problematic precedence for future research involving animals. However, I assume that it is not in the power of the authors to change legal requirements in Sweden.

Reviewer #2: This study comprehensively identified about 5.5 kb-region as endothelial regulatory module involved in blood pressure control based on racehorse exercise performance and blood pressure measurements, and the human HiCap data and transcription factor analysis.

This manuscript has been revised multiple times, with revisions noted by this reviewer. Therefore, this reviewer considers the current manuscript to be suitable for publication.

However, this reviewer requests comments or revisions on the following:

Comments:

In this study, elite haplotypes showed a high frequency in CBT and SB, but a low frequency of 0.07 in TB. Since TB is a racehorse, it may be expected that the frequency of this haplotype would be higher as the identified region is associated with racing ability. Why isn't that the case?

Is it due to the difference in thoroughbred and trotter racing? Or does this allele not function significantly in thoroughbreds? This reviewer was concerned because the title stated "with elite athletic performance."

Line 389: myostatin?

Reviewer #3: please see attached

**Have all data underlying the figures and results presented in the manuscript been provided?**

Reviewer #1: Yes

Reviewer #2: Yes

Reviewer #3: Yes

PLOS authors have the option to publish the peer review history of their article (what does this mean?). If published, this will include your full peer review and any attached files.

Reviewer #1: No

Reviewer #2: No

Reviewer #3: No

---

## [Editor Report · Decision Letter 2]

15 Apr 2024

Dear Dr Fegraeus,

Thank you very much for submitting your Research Article entitled 'An endothelial regulatory module links blood pressure regulation with elite athletic performance' to PLOS Genetics.

The manuscript was fully evaluated at the editorial level and by independent peer reviewers. The reviewers appreciated the attention to an important topic but identified some concerns that we ask you address in a revised manuscript.

We therefore ask you to modify the manuscript according to the review recommendations. Your revisions should address the specific points made by each reviewer.

Yours sincerely,

Carrie J Finno, DVM, PhD

Guest Editor

PLOS Genetics

Gregory Barsh

Section Editor

PLOS Genetics

---

## [Editor Report · Decision Letter 3]

2 May 2024

Dear Dr Fegraeus,

We are pleased to inform you that your manuscript entitled "An endothelial regulatory module links blood pressure regulation with elite athletic performance" has been editorially accepted for publication in PLOS Genetics. Congratulations!

Yours sincerely,

Carrie J Finno, DVM, PhD

Guest Editor

PLOS Genetics

Gregory Barsh

Section Editor

PLOS Genetics

Comments from the reviewers (if applicable):

Thank you for revising this manuscript based on the reviewers' comments.

**Data Deposition**

http://datadryad.org/submit?journalID=pgenetics&manu=PGENETICS-D-23-00898R3

**Press Queries**

---

## [Editor Report · Acceptance letter]

16 May 2024

PGENETICS-D-23-00898R3 

An endothelial regulatory module links blood pressure regulation with elite athletic performance 

Dear Dr Fegraeus, 

We are pleased to inform you that your manuscript entitled "An endothelial regulatory module links blood pressure regulation with elite athletic performance" has been formally accepted for publication in PLOS Genetics! Your manuscript is now with our production department and you will be notified of the publication date in due course.

With kind regards,

Anita Estes

PLOS Genetics

On behalf of:
